# Integrin β1/Cell Surface GRP78 Complex Regulates TGFβ1 and Its Profibrotic Effects in Response to High Glucose

**DOI:** 10.3390/biomedicines10092247

**Published:** 2022-09-10

**Authors:** Jackie Trink, Renzhong Li, Evan Squire, Kian O’Neil, Phoebe Zheng, Bo Gao, Joan C. Krepinsky

**Affiliations:** Division of Nephrology, McMaster University, St. Joseph’s Healthcare, Hamilton, ON L8N 4A6, Canada

**Keywords:** cell surface GRP78, integrin β1, TGFβ1, diabetic kidney disease, cell signaling, fibrosis, mesangial cell

## Abstract

Diabetic kidney disease (DKD) is the leading cause of kidney failure worldwide. Characterized by overproduction and accumulation of extracellular matrix (ECM) proteins, glomerular sclerosis is its earliest manifestation. High glucose (HG) plays a central role by increasing matrix production by glomerular mesangial cells (MC). We previously showed that HG induces translocation of GRP78 from the endoplasmic reticulum to the cell surface (csGRP78), where it acts as a signaling molecule to promote intracellular profibrotic FAK/Akt activation. Here, we identify integrin β1 as a key transmembrane signaling partner for csGRP78. We show that it is required for csGRP78-regulated FAK/Akt activation in response to HG, as well as downstream production, secretion and activity of the well characterized profibrotic cytokine transforming growth factor β1 (TGFβ1). Intriguingly, integrin β1 also itself promotes csGRP78 translocation. Furthermore, integrin β1 effects on cytoskeletal organization are not required for its function in csGRP78 translocation and signaling. These data together support an important pathologic role for csGRP78/integrin β1 in mediating key profibrotic responses to HG in kidney cells. Inhibition of their interaction will be further evaluated as a therapeutic target to limit fibrosis progression in DKD.

## 1. Introduction

Diabetic kidney disease (DKD) continues to be the leading cause of kidney failure worldwide [1]. The current standard of care for DKD includes glucose and blood pressure optimization as well as the use of renin–angiotensin–aldosterone system (RAAS) blockers and more recently the use of sodium glucose cotransporter 2 (SGLT2) inhibitors for type two diabetics [2,3]. Although DKD progression is slowed by these treatments, many patients still develop kidney failure and require costly life-sustaining therapies including dialysis or kidney transplantation. Not only does this impact quality of life, it also places a significant financial burden on healthcare infrastructure [1]. Hence, there is a major need to identify novel therapeutic targets to prevent DKD progression.

The earliest manifestations of DKD are observed in the glomerulus or filtering unit of the kidney. Here, glomerular basement membrane thickening and mesangial matrix expansion due to the production of extracellular matrix (ECM) proteins by mesangial cells (MC) are key pathological changes that lead to reduction in filtering capacity and DKD progression [4,5,6]. Thus, inhibiting the production of ECM by MCs is a focal point for novel therapeutic treatments. Recently, our lab identified a high glucose (HG)-induced cell surface receptor on MCs, cell surface GRP78 (csGRP78), as an important regulator of PI3K/Akt activation and downstream ECM production [7,8].

Endogenously, GRP78 is found in the endoplasmic reticulum (ER), where it is responsible for maintaining homeostasis and proper protein folding [9]. However, under conditions of ER stress including HG, its translocation to the cell surface in association with the co-chaperone MTJ1 has been described [10,11]. At the cell surface, GRP78 can be activated by ligands such as activated alpha 2-macroglobulin (α2M*) to promote activation of intracellular signaling cascades. These may differ depending on ligand identity and the nature of its binding to csGRP78 [10]. In MCs, we have shown that α2M* is increased by HG and its binding to csGRP78 is important for activation of PI3K/Akt signaling and matrix upregulation.

In keeping with the lack of an identified transmembrane domain, GRP78 is known to reside as a peripheral protein on the plasma membrane with all regions exposed on the cell surface [12]. Given its cell surface localization, GRP78 association with a transmembrane protein “coreceptor” is required for intracellular signaling. These are cell and context dependent [10,12]. We have recently shown that csGRP78 interacts with the transmembrane receptor integrin β1 under HG conditions [7]. Inhibition of either csGRP78 or integrin β1 attenuated both FAK and Akt activation by HG, suggesting that integrin β1 is a functional coreceptor for csGRP78 in mediating this profibrotic pathway [7]. Furthermore, previous studies have shown that PI3K/Akt signaling promotes synthesis of the well-characterized profibrotic cytokine transforming growth factor β1 (TGFβ1) [13,14,15], known to be a key pathogenic factor in DKD [16]. In this study, we determine whether integrin β1 functions as a required coreceptor for csGRP78 profibrotic signaling in response to HG in MCs, and the relevance of the csGRP78/integrin β1 axis as a pathologic regulator of TGFβ1 production. Findings will provide insight into its potential as an anti-fibrotic therapeutic target for DKD. 

## 2. Materials and Methods

### 2.1. Cell Culture, Reagents and Protein Extraction

Primary MCs from C57BL/6 mice were previously isolated using Dynabeads. Cells were cultured in DMEM with 5.6 mM glucose supplemented with 16% FBS, 100 µg/mL streptomycin/penicillin and grown at 37 °C in 95% O_2_, 5% CO_2_. MCs were serum deprived with 0.5% FBS for 24 h prior to treatment with HG (30 mM) with or without csGRP78 inhibitors: Subtilase cytotoxin A (SubA, generously donated by Dr. J. Paton, University of New Mexico School of Medicine) (25 ng/mL) [17] or antibodies targeting the N- or C-terminus of GRP78 (N88, C38, respectively, gifted by Dr. S. Pizzo, Duke University Medical Centre at Durham) or pre-adsorbed IgG control (10 µg/mL) [18]. 

The following antibodies were used: pFAK Tyr397 (1:1000, Cell Signaling, Danvers, MA, USA), total FAK (1:1000, Cell Signaling), pSmad3 Ser423/425 (1:4000, Novus, St. Charles, MO, USA), total Smad3 (1:1000, Abcam, Cambridge, UK), latency-associated peptide (LAP)-TGFβ1 (1:1000, R&D Systems, Minneapolis, MN, USA), fibronectin (FN) (1:1000, Novus), Collagen IV (Col IV) (1:1000, Cell Signaling), α-tubulin (1:40,000, Sigma, St. Louis, MO, USA), GRP78 (1:1000 BD Biosciences, San Jose, CA, USA), platelet-derived growth factor receptor β (PDGFRβ) (1:1000, Santa Cruz Biotechnology, Dallas, TX, USA), integrin β1 (1:1000, Abcam), integrin β1 activating antibody (P4G11) (Sigma) and LEAF^TM^ purified anti-mouse/rat active integrin β1 neutralizing antibody (Biolegend, San Diego, CA, USA). The following inhibitors were used: FAK inhibitor (PF573228, 2 µM), cytochalasin D (200 ng/mL), latrunculin B (400 nM), and colchicine (5 µg/mL). Protein harvest from whole cell lysates was described previously [19]. Proteins were separated using SDS-PAGE and immunoblotted to assess protein expression.

### 2.2. Transfection and Luciferase 

For short interfering (si)RNA and luciferase experiments, MCs were plated at 50% confluence and transfected with 100 nM of MTJ1, integrin β1 or control siRNA (Silencer Select, Thermo Fisher, Waltham, MA, USA) using Lipofectamine (Invitrogen, Waltham, MA, USA) or 1 µg of the Smad3-responsive luciferase reporter construct CAGA12-luciferase (donated by Dr. M. Bilandzic, Hudson Institute of Medical Research, Clayton, Australia) with 0.05 µg pCMV β-galactosidase (β-Gal, Clontech, Mountain View, CA, USA) using Effectene (Qiagen, Hilden, Germany), respectively. The medium was changed 18 h after transfection and cells serum deprived prior to treatment. 

Electroporation was used to transfect cells with pcDNA3.1 GRP78 ∆KDEL (GRP78 lacking the KDEL domain which localizes it to the ER, thus enabling significant localization to the cell surface [20]). Empty vector was used as a control. Confluent MCs were trypsinized and centrifuged in 20% FBS DMEM without antibiotics. Cells (200 µL, 5 × 10^5^/well) were electroporated in a 4 mm gap cuvette with 10 µg plasmid for one 30 ms pulse at 250 V (ECM 399, BTX Harvard Apparatus, Holliston, MA, USA) before replating. MCs were then serum-deprived, treated and harvested as above.

For luciferase harvest, cells were lysed in 1× Reporter Lysis Buffer (Promega, Madison, WI, USA) and stored at −80 °C overnight. Luciferase activity was measured after clarification of lysates using the Luciferase Assay System (Promega) with a luminometer (Junior LB 9509, Berthold, Bad Wildbad, Germany). β-Gal activity was used to normalize samples and was measured using the β-Galactosidase Enzyme Assay System (Promega) with a SpectraMax Plus 384 Microplate Reader (Molecular Devices, San Jose, CA, USA) set to read absorbance at 420 nm.

### 2.3. RNA Extraction and qPCR

Trizol (Invitrogen) was used to extract RNA and 1 µg was reverse transcribed using qScript Supermix Reagent (Quanta Biosciences). Using Power SYBR Green PCR Master Mix on the Applied Biosystems Vii 7 Real-Time PCR System, quantitative PCR was performed to determine changes in mRNA expression relative to 18 S using the ΔΔCt method with the following primers: TGFβ1 forward 5′-AAACGGAAGCGCATCGAA-3′ and reverse 5′-GGGACTGGCGAGCCTTAGTT-3′ and 18S forward 5′-GCCGCTAGAGGTGAAATTCTTG-3′ and reverse 5′-CATTCTTGGCAAATGCTTTCG-3′.

### 2.4. Biotinylation

After treatment, MCs were incubated with 1 mg/mL EZ-linked Sulfo-Biotin (Pierce, Waltham, MA, USA) for 30 min. Excess Sulfo-Biotin was removed by washes with 0.1 M glycine in PBS, then cells were lysed, clarified, and normalized concentrations of proteins were incubated overnight with a 50% Neutravidin slurry (Fisher) to capture biotin-tagged proteins. The following day, beads were washed with lysis buffer 5 times, after which tagged proteins were cleaved from beads by boiling for 10 min in 2× PSB. Samples were separated using SDS-PAGE prior to immunoblotting.

### 2.5. TGFβ1 ELISA

Conditioned medium was collected from MCs after treatment and total secreted TGFβ1 was measured after acid activation using the TGFβ1 Quantikine ELISA kit (R&D Systems). 

### 2.6. TGFβ1 Bioassay with Mink Lung Epithelial Cells (MLECs)

MLEC stably transfected with the PAI-1 luciferase promoter construct were used. They were cocultured with MCs in MEM with 10% FBS, plated at 5000 and 25,000 cells/well, respectively, on a 12-well plate (1:5 ratio MLEC:MC). The next day, cells were serum deprived for 18 h followed by treatment with HG and inhibitors. At collection, cells were lysed in 1× Reporter Lysis Buffer (Promega) and stored at −80 °C overnight before analysis of PAI-1 luciferase activity.

### 2.7. Surface Protein Co-Immunoprecipitation from Live Cells

Cells were washed three times with PBS and then incubated in starvation medium with 5 µg anti-GRP78 N88, anti-integrin β1 or control IgG at 4 °C for 2 h with gentle agitation. MCs were then washed with PBS, lysed, and passed through a 25-gauge needle (Precision Glide Needle, BD, San Jose, CA, USA) 5 times to ensure complete lysis. Lysates were clarified and Protein G beads (rProtein G Agarose, Invitrogen) were added to normalized samples for 2 h at 4 °C on a rocking plate. Beads were washed with lysis buffer and proteins then eluted by boiling for 10 min in 2× PSB. Samples were analyzed using SDS-PAGE and immunoblotting. 

### 2.8. Statistical Analysis

For comparison between two or more groups, a two-tailed *t*-test or one-way ANOVA were used, respectively, using GraphPad Prism 6.0. Tukey’s post hoc analysis was completed when more than two groups were analyzed. Statistical significance was designated at *p* < 0.05 and data are presented as the mean ± SEM.

## 3. Results

### 3.1. csGRP78 Mediates TGFβ1 Transcription in Response to HG

To assess the role of csGRP78 in the regulation of TGFβ1 transcription, we first confirmed the increase in surface expression of GRP78 under HG treatment (Figure 1A). Next, we assessed the effects of csGRP78 inhibition on HG-induced TGFβ1 transcript upregulation. We first used an antibody that targets the C-terminus of GRP78, termed C38 [18]. We previously showed that the C-terminal targeting antibody C-20, no longer manufactured, inhibits HG-induced FAK/Akt activation [7], and confirmed similar efficacy of C38 (not shown). Figure 1B shows that C38 prevented HG-induced TGFβ1 transcript upregulation. We next used the enzyme SubA, a cell-impermeable proteinase that selectively cleaves the C-terminus of cell surface GRP78 [17], which we showed also inhibits HG-induced signaling [7]. This similarly prevented HG-induced TGFβ1 transcript upregulation (Figure 1C). Finally, using siRNA we downregulated the co-chaperone MTJ1, required for HG-induced translocation of GRP78 to the cell surface [7,21]. This also prevented TGFβ1 mRNA upregulation induced by HG (Figure 1D). We further assessed if HG-induced csGRP78 regulates TGFβ1 promoter activity in MCs. Figure 1E–G show that TGFβ1 promoter luciferase activity in HG was significantly reduced with csGRP78 inhibition using C38, SubA and MTJ1 knockdown, respectively. These data support a role for csGRP78 in regulating TGFβ1 transcription.

### 3.2. HG-Induced TGFβ1 Protein Synthesis and Secretion Are Mediated by csGRP78

We next wished to assess whether csGRP78 also modulates TGFβ1 production and secretion in HG. We first measured protein expression of the secreted inactive form of TGFβ1, LAP-TGFβ1. In Figure 2A,B, we observed cellular inhibition of HG-induced LAP-TGFβ1 expression by C38 or SubA, respectively. Further, knockdown of MTJ1 also abrogated LAP-TGFβ1 upregulation (Figure 2C). Next, we analyzed the secretion of total TGFβ1 by ELISA. Again, we observed a significant reduction in secreted TGFβ1 with C38 (Figure 2D), SubA (Figure 2E), or after downregulation of MTJ1 with siRNA (Figure 2F), implicating csGRP78 in the modulation of TGFβ1 protein synthesis and secretion.

### 3.3. csGRP78 Facilitates HG-Induced Smad3 Activation

As our data thus far implicate a role for csGRP78 in the production and secretion of TGFβ1, we next wished to confirm that downstream signaling was also affected. We thus assessed activation of Smad3 by measuring its phosphorylation (at C-terminus Ser473/475) and activation of the Smad3-responsive reporter CAGA12 luciferase [16,22]. In Figure 3A,B, we observed attenuation of HG-induced Smad3 phosphorylation by C38 and SubA, respectively. Further, HG-induced CAGA12 luciferase activity was also inhibited by C38 (Figure 3C) and SubA (Figure 3D). These data clearly show that csGRP78 mediates downstream TGFβ1 signaling in MCs in response to HG.

### 3.4. Integrin β1 Interaction with csGRP78 Is Required for TGFβ1 Upregulation and Signaling in HG 

Our lab has previously shown that integrin β1 may play an important role in regulating csGRP78-mediated profibrotic FAK/Akt activation [7], with Akt known to regulate TGFβ1 induction by HG [15]. We thus next wished to investigate if integrin β1 was also required for modulating TGFβ1 upregulation and activity in HG. Using a neutralizing antibody targeting the conformationally active form of integrin β1, we assessed whether inhibition of integrin β1 would affect TGFβ1 production, secretion and downstream activity. In Figure 4A and B, we observed attenuation of HG-induced TGFβ1 production with integrin β1 neutralization at both the transcript and protein level, respectively. HG-induced secretion of total TGFβ1 (Figure 4C) and its downstream activity measured by phosphorylation of Smad3 (Figure 4D) and CAGA12 luciferase activity (Figure 4E) were also inhibited by integrin β1 neutralization. Further, knockdown of integrin β1 prevented HG-induced FAK activation, TGFβ1 upregulation and downstream TGFβ1 signaling as well as ECM protein (fibronectin, collagen IV) upregulation (Figure 4F). 

We next assessed whether GRP78 associates with both integrin β1 and LAP-TGFβ1 at the cell surface. Cell surface GRP78 was immunoprecipitated from live cells after treatment as described in methods. In Figure 4G, we observed HG-induced association of csGRP78 and integrin β1 as we have shown previously [7], in addition to csGRP78 association with LAP-TGFβ1. In Figure 4H, reverse immunoprecipitation with cell surface integrin β1 confirmed this association under HG treatment. Importantly, association between these proteins was attenuated by csGRP78 inhibition with the C38 antibody. We next sought to confirm that both integrin β1 and csGRP78 are required for TGFβ1 signaling in HG. For this assay, MCs were cocultured with MLEC stably transfected with the Smad3-dependent PAI-1 promoter luciferase construct, allowing for assessment of TGFβ1 activity. After plating, cells were treated with HG and inhibitors of either csGRP78 or integrin β1, or control IgG. As shown in Figure 4I, HG led to TGFβ1 activation as showed by increased luciferase activity in cell lysate. This was inhibited by both C38 and the integrin β1 neutralizing antibody. These data implicate a direct role for both csGRP78 and integrin β1 in modulating TGFβ1 activation under HG. 

### 3.5. Overexpression of csGRP78 Augments TGFβ1 Synthesis, Secretion and Downstream Profibrotic Signaling

We previously showed that overexpression of GRP78 lacking the ER-retention sequence KDEL (ΔKDEL) localizes it to the cell surface and augments HG-induced ECM and profibrotic cytokine production [8]. Here, we aimed to investigate whether the overexpression of csGRP78 could also augment TGFβ1 production, secretion and downstream signaling, and whether this required integrin β1. In Figure 5A, we observed an increase in the basal expression level of LAP-TGFβ1 in MCs transfected with ΔKDEL, and this response was augmented by HG. Similarly, we observed an increase in TGFβ1 secretion (Figure 5B) and phosphorylation of Smad3 (Figure 5C) with ΔKDEL alone, both of which were further augmented by HG. We next wished to evaluate if integrin β1 was required for this signaling cascade. We thus treated ΔKDEL-overexpressing MCs with the integrin β1 neutralizing antibody. Figure 5D and E show that integrin β1 neutralization reduced expression of LAP-TGFβ1 and pSmad3 to levels seen in cells transfected with the empty control vector, indicating a critical role for integrin β1 in csGRP78 signaling.

### 3.6. Integrin β1 Contributes to GRP78 Cell Surface Translocation

Previous studies have shown manganese (Mn) to be a non-specific integrin activator, with its activation of FAK downstream of integrins also confirmed [23,24]. We thus initially used Mn to determine whether integrin activation was necessary or sufficient to induce GRP78 localization to the cell surface as well as for TGFβ1 production and signaling. In Figure 6A, we observed significant Mn-induced translocation of GRP78 to the cell surface. In Figure 6B,C, both TGFβ1 production and downstream signaling (measured by Smad3 phosphorylation) were increased by Mn treatment. Importantly, both csGRP78 and integrin β1 are required for this as inhibition of either csGRP78 (Figure 6D) or integrin β1 (Figure 6E) resulted in a loss of Mn-induced Smad3 phosphorylation. Using an antibody that specifically activates integrin β1 (P4G11), we confirmed that localization of GRP78 to the cell surface (Figure 6F) and activation of downstream signaling measured as FAK and Smad3 phosphorylation (Figure 6G and 6H, respectively) were induced by integrin β1. Furthermore, these were prevented by csGRP78 inhibition (Figure 6I,J). Finally, we tested whether integrin β1 was needed for GRP78 translocation to the cell surface in response to HG. As shown in Figure 6K, integrin β1 downregulation with siRNA inhibited HG-induced translocation. Taken together, these data show the critical interdependence of csGRP78 and integrin β1 in the profibrotic response to HG. 

### 3.7. HG-Induced csGRP78 Translocation Is Independent of Cytoskeleton Organization

The importance of integrins in the regulation of cytoskeleton organization, in addition to their role as modulators of transmembrane signaling pathways, is well established [25,26,27,28]. Having shown that integrin β1 is required for csGRP78 signaling in HG, we next sought to determine whether cytoskeletal integrity is required for csGRP78 signaling. We first assessed the actin cytoskeleton. Phalloidin staining for actin shown in Figure 7A confirms that the inhibitors cytochalasin D and latrunculin B both disrupted the actin cytoskeleton. However, neither prevented HG-induced translocation of GRP78 to the cell surface (Figure 7B). We then tested whether the microtubule network was required. Its disruption with colchicine was first confirmed by staining for tubulin (Figure 7C), but it also did not reduce HG-induced csGRP78 expression (Figure 7D), suggesting that cytoskeletal integrity is not required for GRP78 surface translocation. 

We next sought to determine whether activity of FAK downstream of integrin β1, which regulates signaling events as well as cytoskeletal organization [27], was required for signaling in HG. In Figure 7E, we observed a loss of HG-induced cell surface presentation of GRP78 with a FAK inhibitor. FAK inhibition also resulted in a loss of HG-induced TGFβ1 production and signaling as well as downstream reduction in ECM protein (fibronectin, collagen IV) synthesis (Figure 7F). Together, these data support an important signaling function for integrin β1, in cooperation with csGRP78, to promote the profibrotic response to HG in MCs.

## 4. Discussion

Recently, we identified csGRP78 as an important regulator of profibrotic signaling in MCs under HG conditions and showed the de novo expression of GRP78 at the cell surface in vivo in models of DKD [7,8]. We also implicated activated alpha 2-macroglobulin (α2M*) as a ligand for csGRP78, which is required for its profibrotic signaling [7,8]. However, as csGRP78 is found completely extracellularly, a transmembrane coreceptor is required to transmit signaling intracellularly. While several such proteins have been shown to associate with csGRP78 [10], the identity of the coreceptor required for HG signal transduction is as yet unknown. In this study, we built on our previous work in which we showed that csGRP78 interacts with integrin β1 under HG conditions, suggesting that this integrin could be the potential binding partner relevant for signaling [7]. Here, we showed a critical role for integrin β1 in transmitting profibrotic FAK/Akt signaling by csGRP78 in response to HG and demonstrated that integrin β1/csGRP78 are required for HG-induced TGFβ1 production, secretion and downstream signaling in MCs. Intriguingly, we also showed that integrin β1 activation, independent of its role in cytoskeletal organization, can induce GRP78 translocation to the cell surface, thereby potentially amplifying profibrotic signaling. These data, summarized in Figure 8, support further evaluation of the integrin β1/csGRP78 signaling axis as a potential therapeutic target for DKD. 

Previous studies have shown the importance of integrin β1 in regulating the assembly and production of ECM proteins that contribute to the fibrotic phenotype seen in DKD [29,30]. Integrin β1 was also shown to bind to LAP and through traction release the mature profibrotic cytokine, indirectly contributing to the overproduction of ECM proteins [31]. Our MLEC coculture data support a role for integrin β1 in the activation of TGFβ1, showing a decrease in TGFβ1 functional effects in the presence of either integrin β1 or csGRP78 blockade. That both reduce TGFβ1 effects in this functional assay to the same degree and to baseline levels supports their integrated and cooperative role. The molecular mechanism underlying the role of csGRP78 in regulating extracellular TGFβ1 activation, however, requires further elucidation. Our immunoprecipitation data show interaction between integrin β1, LAP-TGFβ1 and csGRP78 in response to HG. It is possible that csGRP78 facilitates interaction with ECM proteins required to provide traction for release of TGFβ1 [32], such as with the matricellular ECM glycoprotein thrombospondin-1 [33,34,35]. Further studies are required to address this possibility. 

Our data also add to these known roles of integrin β1 in ECM accumulation. We show that, in cooperation with csGRP78, integrin β1 also enables the increased synthesis of TGFβ1, thereby amplifying profibrotic capability. This is important since the direct targeting of TGFβ1 is not clinically feasible due to the pleiotropic effects of the cytokine and associated adverse effects with its inhibition [36]. Indirect methods relevant to disease pathology, such as targeting the integrin β1-csGRP78 interaction, may thus provide greater clinical benefit without adverse effects.

Integrins exist as heterodimeric transmembrane glycoproteins that consist of non-covalently associated α and β subunits [27]. While our studies assessed the role of integrin β1 as a csGRP78 signaling partner, we have not as yet identified its relevant alpha binding partner, with 18 alpha subunits known to exist in mammals. Indeed, each αβ heterodimer has been shown to have different affinities for their ligand ECM components, and integrin heterodimers that bind to the same ligand can further elicit distinct signaling cascades [28,37]. Whether there is a disease-specific β1 heterodimer that interacts with csGRP78 to propagate the profibrotic signaling pathway we observed in this study requires clarification. Furthermore, given the various distinct cell types in the kidney, whether integrin β1-csGRP78 signaling is more broadly seen, such as in tubular cells or fibroblasts which contribute to the tubulointerstitial fibrosis that defines later stages of disease, requires further study. Similarly, whether the alpha binding partner is consistent across cell types in the diabetic kidney also needs to be identified. 

Previous studies have implicated several integrin heterodimers in the pathogenesis of DKD and other kidney diseases. Integrin α2β1 has been shown to mediate glomerular fibrosis in kidney disease through the production of collagen IV and reactive oxygen species. Deletion or inhibition of integrin α2β1 showed protective effects to the glomerulus and prevented the development of fibrosis [38]. The fibronectin-assembling integrin α5β1 was shown to be upregulated by TGFβ1 in MCs and by HG in both MCs and podocytes, but has not as yet been directly shown to promote kidney disease [39,40,41]. Inhibition of both integrin αvβ1 and αvβ3 were shown to ameliorate kidney fibrosis in a diabetic mouse and porcine model, respectively [31,42]. Future studies will determine whether these known profibrotic integrin heterodimers function at least in part through csGRP78. 

We previously showed an important role for cytoskeleton organization in mechanical stress-induced profibrotic signaling in MCs, with disruption of the actin or microtubule cytoskeleton inhibiting signaling [19,43]. Our current study, however, has identified a distinct mechanism for integrin β1/FAK signaling that is independent of cytoskeleton integrity in MCs. This may be important therapeutically since inhibition of csGRP78/integrin β1 interaction should not disrupt cytoskeleton regulation. Thus, this important cellular process will remain intact, providing targeted disease specificity which reduces the likelihood for adverse effects. 

Our previous study has shown that α2M*, a high affinity ligand for csGRP78, is increased significantly in diabetic kidneys and is required for profibrotic signaling in response to HG in MCs [8]. How α2M*, csGRP78 and integrin β1 interact at a molecular level is as yet unknown. It is possible that HG may induce integrin β1 anchoring of GRP78 to the cell surface, thereby allowing for its presentation on the surface in a particular topology. This may facilitate the interaction of csGRP78 with α2M*, with consequent promotion of profibrotic signaling. Alternatively, α2M* binding to csGRP78 may alter its conformation and facilitate binding to active integrin β1. Future studies will aim to elucidate the particular mechanism by which this signaling occurs.

Overall, we have shown that both csGRP78 and integrin β1 are required for HG-induced TGFβ1 synthesis, secretion and signaling in MCs. Interruption of the interaction between these cell surface proteins may represent a novel therapeutic target to prevent profibrotic signaling in DKD. In vivo studies to assess the efficacy of inhibiting csGRP78/integrin β1/α2M* signaling will elucidate the viability of inhibiting this axis as a potential therapeutic target for DKD.

## Figures and Tables

**Figure 1 biomedicines-10-02247-f001:**
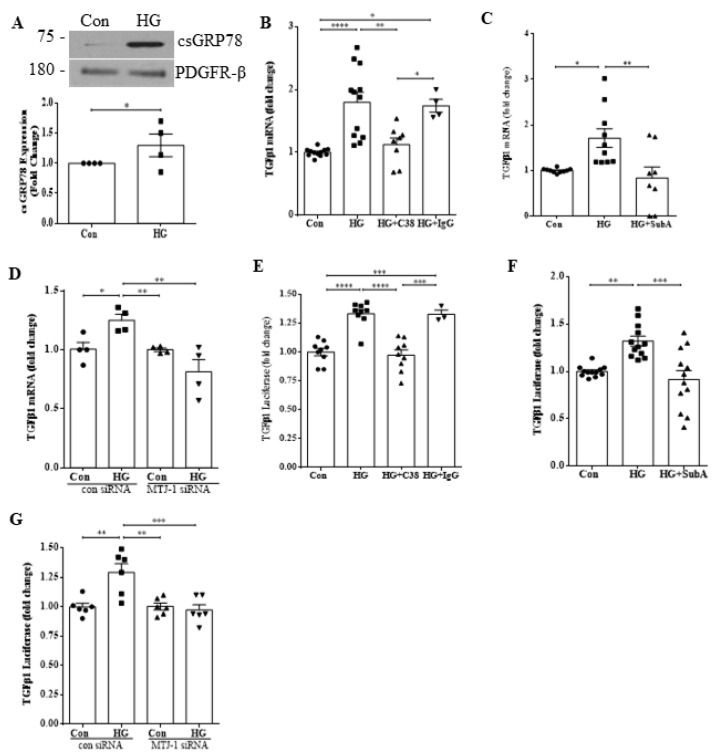
csGRP78 mediates TGFβ1 transcription in response to HG. (**A**) HG (6 h) induced cell surface localization of GRP78 in MCs (*n* = 4, * *p* < 0.05). (**B**) The GRP78 antibody C38 attenuated HG (48 h)-induced TGFβ1 transcript upregulation compared to nonspecific IgG (2 µg for each antibody, *n* = 12, * *p* < 0.05, ** *p* < 0.01, *** *p* < 0.005). (**C**) Cleavage of the C-terminus of csGRP78 with SubA (25 ng/mL) prevented TGFβ1 transcript upregulation by HG (48 h) (*n* = 10, * *p* < 0.05, ** *p* < 0.01). (**D**) siRNA knockdown of MTJ1, required for cell surface translocation of GRP78 in HG, attenuated HG (48 h)-induced TGFβ1 mRNA upregulation compared to control siRNA (100 nM, *n* = 4, * *p* < 0.05 ** *p* < 0.01). HG (48 h)-induced TGFβ1 promoter activity, assessed using a luciferase reporter construct, was also inhibited by (**E**) C38 (*n* = 10, *** *p* < 0.005, **** *p* < 0.0001), (**F**) SubA (*n* = 12, ** *p* < 0.01, *** *p* < 0.005), or (**G**) MTJ1 knockdown (*n* = 6, ** *p* < 0.01, *** *p* < 0.005) in MCs.

**Figure 2 biomedicines-10-02247-f002:**
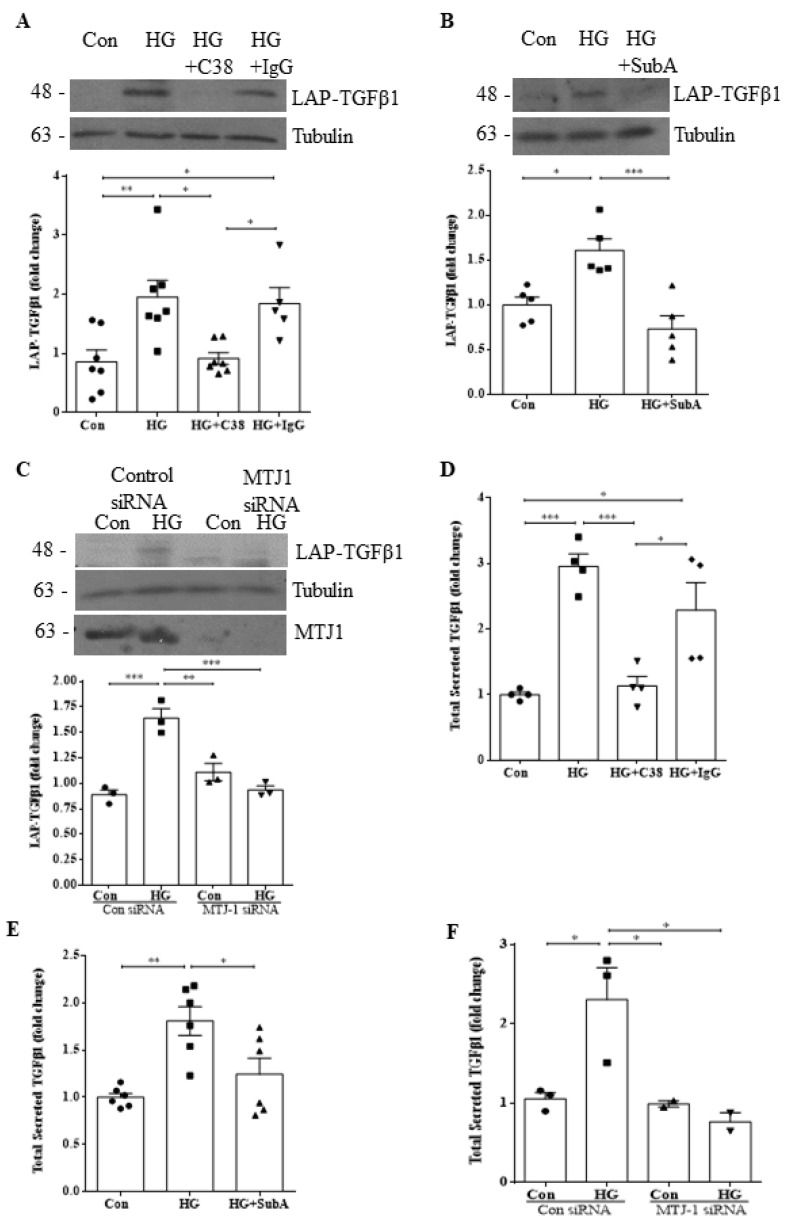
HG-induced TGFβ1 protein synthesis and secretion are mediated by csGRP78. Increased synthesis of LAP-TGFβ1 in response to HG (48 h) was inhibited by (**A**) the GRP78 antibody C38, but not control IgG (2 µg, *n* = 6, * *p* < 0.05, ** *p* < 0.01), **(B**) csGRP78 inhibitor SubA (25 ng/mL, *n* = 5, * *p* < 0.05, *** *p* < 0.005), or (**C**) MTJ1 siRNA, but not control siRNA (100 nM, *n* = 3, ** *p* < 0.01, *** *p* < 0.005). HG (48 h)-induced TGFβ1 secretion, assessed by ELISA of acid-activated medium, was prevented by csGRP78 inhibition using either (**D**) C38 antibody (2 µg, *n* = 4, * *p* < 0.05, *** *p* < 0.005), (**E**) SubA (25 ng/mL, *n* = 6, * *p* < 0.05, ** *p* < 0.01), or (**F**) MTJ1 siRNA knockdown (100 nM, *n* = 3, * *p* < 0.05).

**Figure 3 biomedicines-10-02247-f003:**
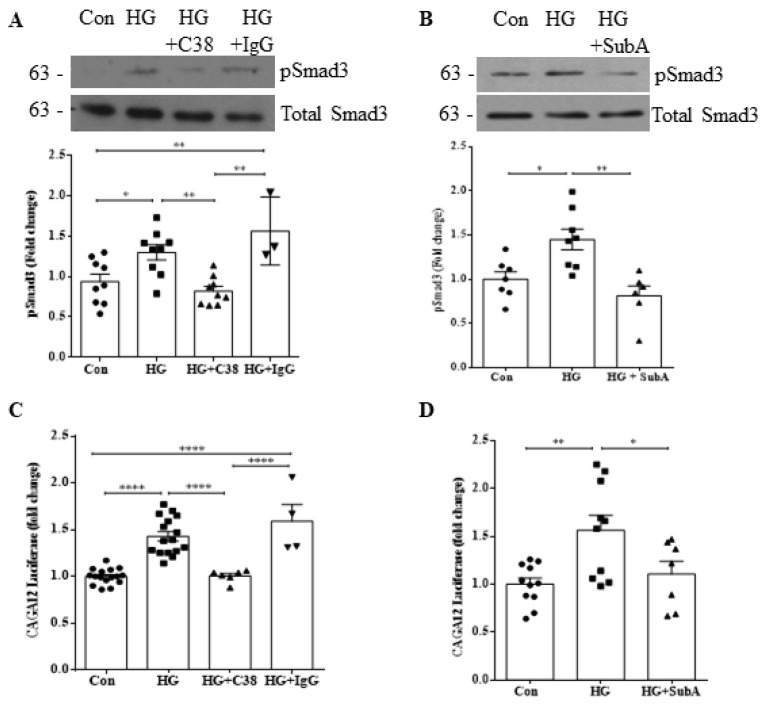
csGRP78 facilitates HG-induced Smad3 activation. HG (48 h)-induced Smad3 activation, assessed by its C-terminal phosphorylation at Ser473/475, was prevented by csGRP78 inhibition with (**A**) C38 antibody, but not nonspecific IgG (2 µg, *n* = 9, * *p* < 0.05, ** *p* < 0.01) or (**B**) SubA (25 ng/mL, *n* = 8, * *p* < 0.05, ** *p* < 0.01). HG (48 h)-increased Smad3 transcriptional activity, assessed using the Smad3-responsive CAGA12 luciferase reporter, was also prevented by csGRP78 inhibition with (**C**) C38 (*n* = 16) or (**D**) SubA (*n* = 12, * *p* < 0.05, ** *p* < 0.01, **** *p* < 0.0001).

**Figure 4 biomedicines-10-02247-f004:**
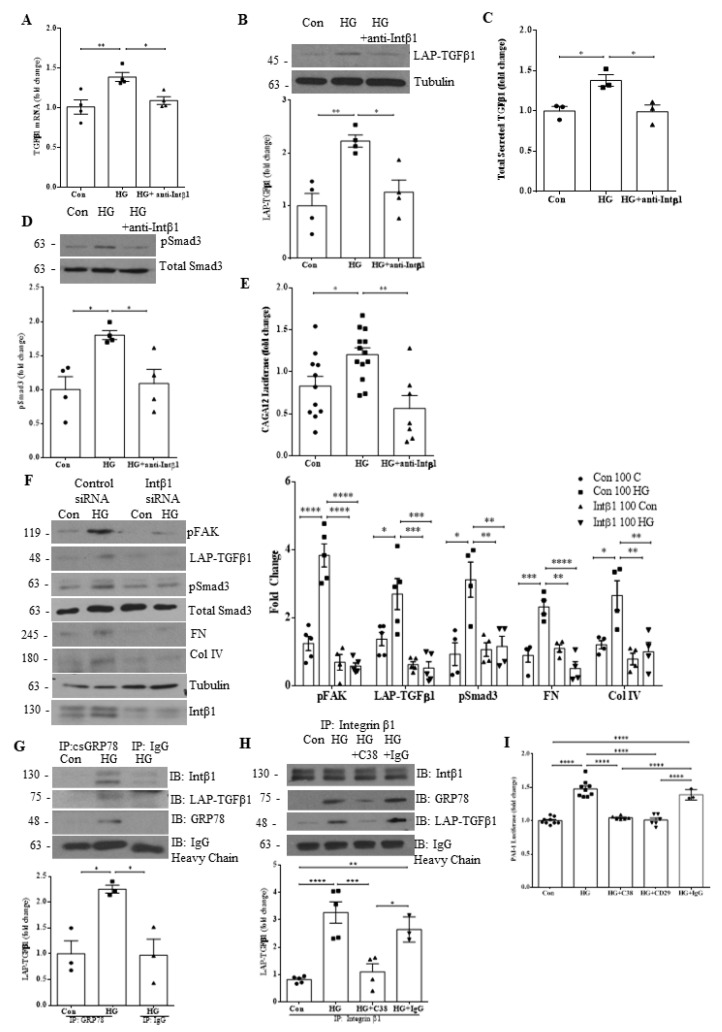
Integrin β1 interaction with csGRP78 is required for TGFβ1 upregulation and signaling in HG. HG-induced TGFβ1 (**A**) transcript upregulation (*n* = 4), (**B**) protein synthesis (*n* = 4), (**C**) secretion (*n* = 3) and signaling measured as (**D**) Smad3 activation (phosphorylation at Ser473/475) (*n* = 4) or (**E**) CAGA12 luciferase activity (*n* = 12) were inhibited by a neutralizing integrin β1 antibody (10 µg, HG 24h for mRNA and 48 h for all others, * *p* < 0.05, ** *p* < 0.01). (**F**) These were also prevented by downregulation of integrin β1 using siRNA, which also inhibited activation of the integrin-regulated kinase FAK (phosphorylation at Tyr397) and upregulation of matrix proteins fibronectin (FN) and collagen IV (Col IV) in response to HG (48 h, *n* = 6, * *p* < 0.05, ** *p* < 0.01, *** *p* < 0.005, **** *p* < 0.0001). (**G**) csGRP78 was immunoprecipitated from live cells after HG for 48 h, showing interaction with integrin β1 and LAP-TGFβ1 (*n* = 3, * *p* < 0.05). (**H**) Reverse immunoprecipitation of cell surface integrin β1 confirmed interaction with csGRP78 as well as with LAP-TGFβ1 in response to HG (48 h) (*n* = 4, * *p* < 0.05, ** *p* < 0.01, *** *p* < 0.005, **** *p* < 0.0001). (**I**) MCs were co-cultured with mink lung epithelial cells (MLEC) stably transfected with Smad3-regulated PAI-1 promoter luciferase. The increase in luciferase activity induced by HG (48 h) was prevented by either csGRP78 inhibition with C38 (2 µg) or antibody neutralization of active integrin β1 (10 µg) (*n* = 9, **** *p* < 0.0001).

**Figure 5 biomedicines-10-02247-f005:**
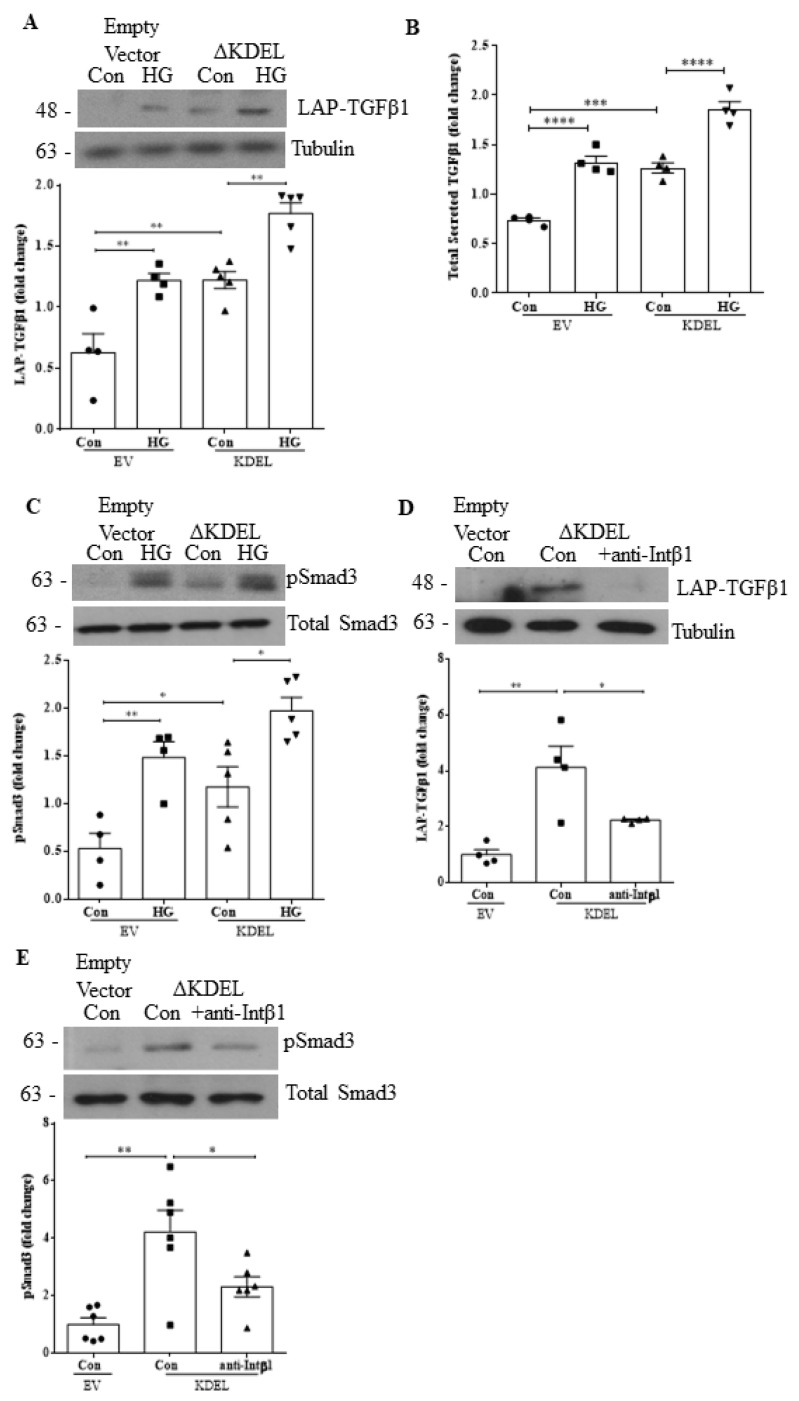
Overexpression of csGRP78 augments TGFβ1 synthesis, secretion, and downstream profibrotic signaling. (**A**) Overexpression of GRP78 ΔKDEL, which increases csGRP78, increased basal LAP-TGFβ1 expression and augmented expression induced by HG (48 h) (*n* = 5, ** *p* < 0.01). Increased basal and HG (48 h)-induced expression of both (**B**) total secreted TGFβ1 (*n* = 4) and (**C**) downstream signaling measured by Smad3 phosphorylation (Ser473/475) (*n* = 5) were also seen (* *p* < 0.05, ** *p* < 0.01, *** *p* < 0.005, **** *p* < 0.0001). GRP78 ΔKDEL-induced increases in (**D**) LAP-TGFβ1 and (**E**) pSmad3 (phosphorylation at Ser473/475) were both attenuated by a neutralizing integrin β1 antibody (10 µg, HG 48 h, *n* = 6, * *p* < 0.05, ** *p* < 0.01).

**Figure 6 biomedicines-10-02247-f006:**
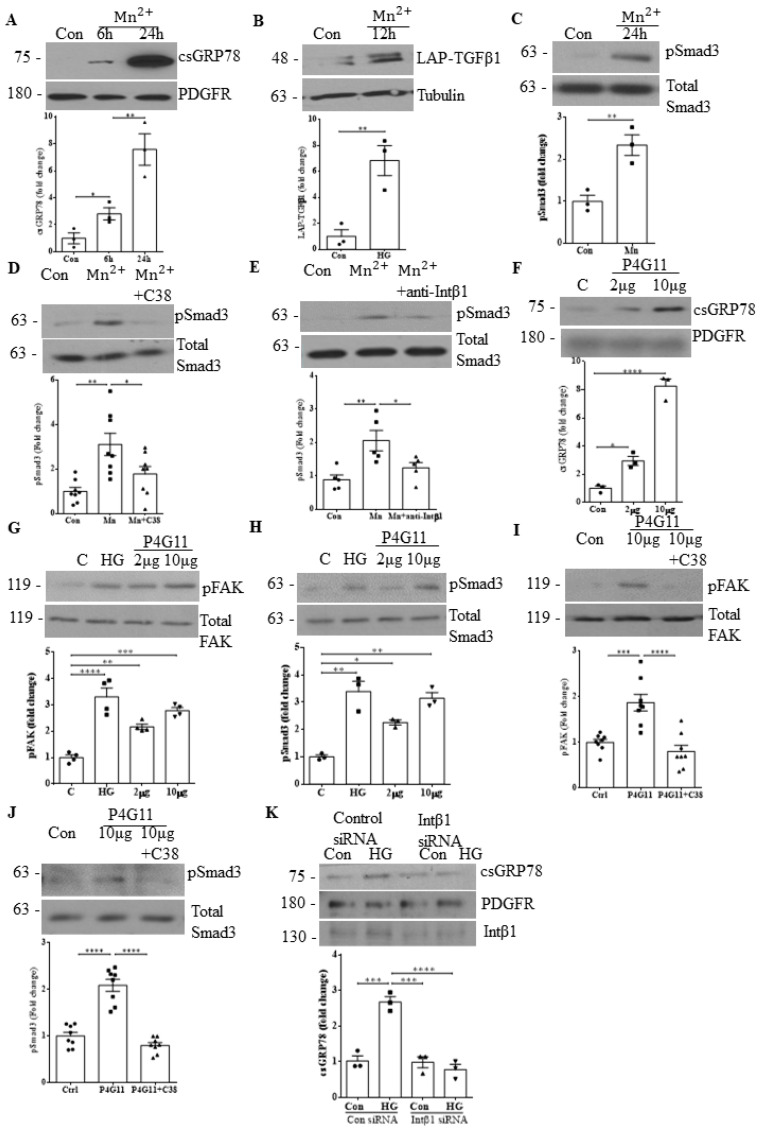
Integrin β1 contributes to GRP78 cell surface translocation. Activation of integrins using manganese. (Mn) induced (**A**) localization of GRP78 to the cell surface (*n* = 3, * *p* < 0.05, ** *p* < 0.01), (**B**) production of LAP-TGFβ1 (*n* = 3, ** *p* < 0.01) and (**C**) downstream TGFβ1 signaling measured as Smad3 phosphorylation (Ser473/475) (*n* = 3, ** *p* < 0.01). Inhibition of either (**D**) csGRP78 using the C38 antibody (*n* = 8, * *p* < 0.05, ** *p* < 0.01) or (**E**) integrin β1 using a neutralizing antibody (*n* = 5, * *p* < 0.05, ** *p* < 0.01) attenuated Mn (24 h)-induced activation of Smad3 (phosphorylation at Ser473/475). To specifically assess the role of integrin β1, its activating antibody P4G11 was used. (**F**) P4G11 (2 or 10 µg, 6 h) increased csGRP78 expression in MCs (*n* = 3, * *p* < 0.05, *** *p* < 0.005). P4G11 (2 or 10 µg, 12 h) also induced activation of both (**G**) FAK (phosphorylation at Tyr397) and (**H**) Smad3 (phosphorylation at Ser473/475) (*n* = 4, ** *p* < 0.01, *** *p* < 0.005, **** *p* < 0.0001). (**I, J**) These were attenuated by csGRP78 inhibition (C38 2 µg, 12 h, *n* = 8, *** *p* < 0.005, **** *p* < 0.0001). (**K**) Downregulation of integrin β1 using siRNA prevented HG (6 h)-induced GRP78 localization to the cell surface in MCs (*n* = 3, *** *p* < 0.005, **** *p* < 0.0001).

**Figure 7 biomedicines-10-02247-f007:**
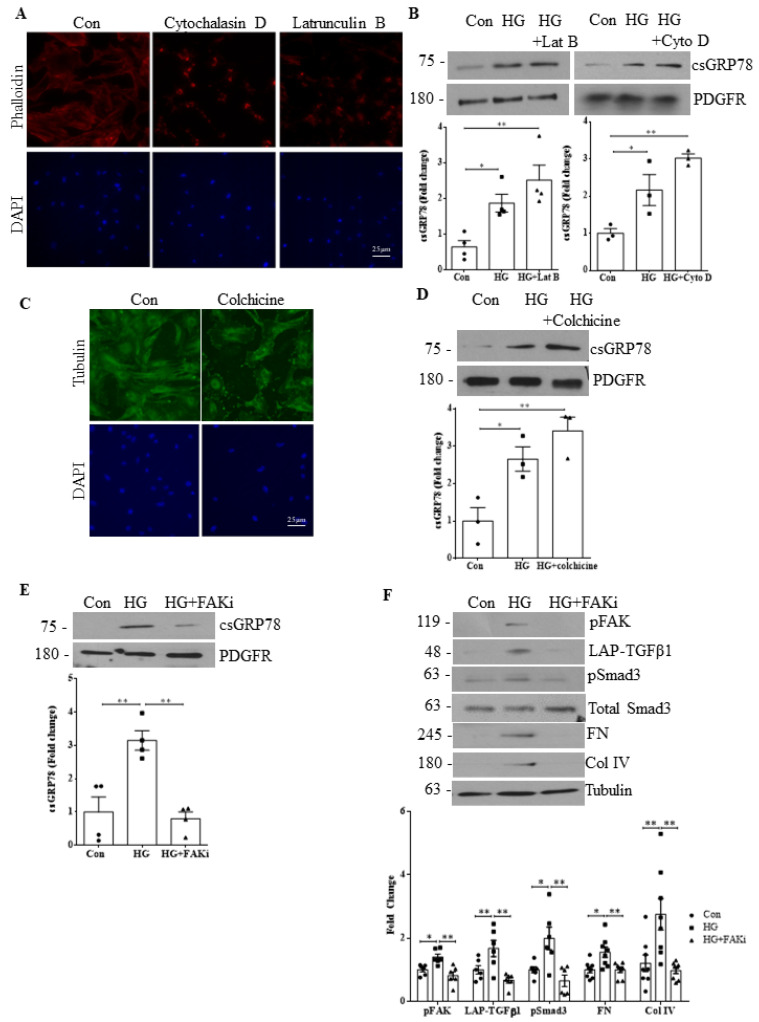
HG-induced csGRP78 translocation is independent of cytoskeleton organization. (**A**) Cells were stained with rhodamine phalloidin to visualize actin filaments, confirming actin cytoskeleton disruption using F-actin inhibitor cytochalasin D (200 ng/mL, 1 h) and G-actin polymerization inhibitor latrunculin B (400nM, 1 h). (**B**) HG (6 h)-induced csGRP78 localization was not affected by either inhibitor (*n* = 4 for Lat B and *n* = 3 for Cyto D, * *p* < 0.05, ** *p* < 0.01). (**C**) Cells were stained with tubulin to visualize microtubule filaments, confirming disruption using colchicine (5 µg/mL). (**D**) Colchicine did not attenuate HG (3 h)-induced localization of GRP78 to the cell surface (*n* = 3, * *p* < 0.05, ** *p* < 0.01). (**E**) Inhibition of FAK, a mediator of integrin β1 signaling, with PF573228 (2µM, FAKi) attenuated HG (6 h)-induced localization of GRP78 to the cell surface (*n* = 3, ** *p* < 0.01). (**F**) HG (48 h)-induced upregulation of LAP-TGFβ1 and downstream activation of Smad3 (phosphorylation at Ser473/475), as well as production of ECM proteins fibronectin (FN) and collagen IV (Col IV) were also prevented by FAK inhibition (*n* = 8, * *p* < 0.05, ** *p* < 0.01).

**Figure 8 biomedicines-10-02247-f008:**
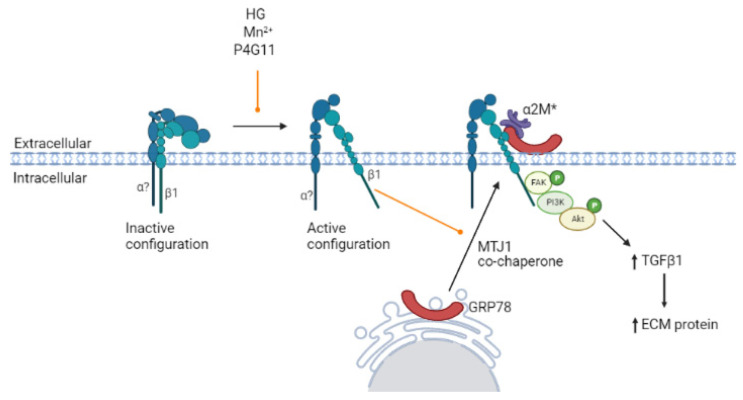
Proposed role of integrin β1 in csGRP78 profibrotic signaling in response to HG in MCs. HG promotes the presentation of GRP78 on the cell surface in MCs (black arrow). Here, interaction with HG-activated integrin β1 induces FAK and downstream Akt signaling. This increases TGFβ1 synthesis, secretion and activation, ultimately leading to an increase in ECM production. Activation of integrins with manganese (Mn) or of integrin β1 more specifically using the activating antibody P4G11 replicates HG effects (orange arrow), supporting a critical role for integrin β1 as a csGRP78 signaling coreceptor. Created with BioRender.

## Data Availability

The data obtained and presented in this article are available from the corresponding author upon reasonable request.

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
