# Peer review of "Integrin β1/Cell Surface GRP78 Complex Regulates TGFβ1 and Its Profibrotic Effects in Response to High Glucose"

_biomedicines, 2022, doi:10.3390/biomedicines10092247_

Round 1

Reviewer 1 Report

Examining the figures vs. the original images I see a great number of very disturbing problems. These have to be addressed specifically and completely before this paper is considered for review.

Major problems.

Fig. 2C. The LAP-TGFB1 and Tubulin images are from 2 different gels. The loading control should use the same gel.

Fig. 3B. The pSMAD3 results from the two replicas (lanes1-3 vs. lanes 4-6) are not the same, and the lanes used for quantitation are different pSMAD3 (lanes 1-3) and Smad3 (lanes 4-6).

Fig. 4H. Beta-integrin and LAP-TGFB1 image in the original westerns do not match those presented in the manuscript.

Fig. 5A and D. The LAP-TGFB1 and Tubulin images are from 2 different gels. The loading control should use the same gel.

Fig. 5E. Fig. 3B. The pSMAD3 results from the two replicas (lanes1-3 vs. lanes 4-6) are not the same, and the lanes used for quantitation are different pSMAD3 (lanes 1-3) and Smad3 (lanes 4-6). However, in this case the incompatibility of the used lanes may not make a difference in the quantitation of the results.

Fig. 6E. The pSMAD3 results from the two replicas (lanes1-3 vs. lanes 4-6) are not the same, and the lanes used for quantitation are different pSMAD3 (lanes 1-3) and Smad3 (lanes 4-6). 

Fig. 6F. PDGFR image in the manuscript does not match or is squeezed compared to the original image.

Fig. 6H, I and J. pSMAD3 and SMAD3 images are from different gels. The results may be from experimental replicas, but it is more appropriate to use the results from the same gel to normalize the data due to potential experimental variations.

Fig. 7E. The csGRP78 and PDGFR are from different gels. The results may be from experimental replicas, but it is required to use the results from the same gel to normalize the data due to potential experimental variations.

Fig. 7F. Westerns use images from mismatched gels or mismatched lanes. The results may be from experimental replicas, but it is more appropriate to use the results from the same gel to normalize the data due to potential experimental variations.

Reviewer 2 Report

The manuscript by Trink and coll. is of good quality. The data presented are convincing and the conclusions are supported by the results. However, a few clarifications are needed.

Materials and methods section 2.1: What is the concentration of glucose in the control condition?

Figure 1: In the figure legend the descriptions of 1F and 1G are lacking.

Figure 1B: Is there a statistical significance between the third condition (HG+C38) and the fourth condition (HG+IgG)? If not, the text should be changed as the data do not demonstrate that C38 prevented HG-induced TGFβ1 transcript upregulation.

Fig1 and 2: it should be interesting to show the efficiency of MJT1 silencing

Fig 5: what is the percentage of cells expressing the GRP78dKDEL construct?

Fig.4 and 7: The authors measured the endogenous expression of FN and CollIV. It seems that they measured the amount of these proteins in cellular extracts, but it should be clarified in the materials and methods section. In our lab, by using skin fibroblasts, we observed that the main part of the endogenously expressed ECM proteins is secreted in the culture medium. Do the authors check the presence of FN and CollIV in the culture medium of mesangial cells? If not, they should explain why the measurements of FN and CollIV in the cellular extracts are sufficient.

Round 2

Reviewer 1 Report

This is an extensive manuscript that builds on the body of work describing the role of csGRP78 in renal fibrosis associated with DKD. The only improvement that is needed is the relabeling of the axis titles in the graphs sine they are not legible.

Author Response

The font size of all axis on graphs has been increased to be more legible in figures.